# Chronobiologically-informed features from CGM data provide unique information for XGBoost prediction of longer-term glycemic dysregulation in 8,000 individuals with type-2 diabetes

**Jamison H. Burks** [1], **Leslie Joe** [2], **Karina Kanjaria** [2], **Carlos Monsivais** [2],
**Kate O'laughlin** [2], **Benjamin L. Smarr** [1,2]*

**1** Shiu Chen – Gene Lay Department of Bioengineering, University of California San Diego, La Jolla, California, United States of America, **2** Halicioğlu Data Science Institute, University of California San Diego, La Jolla, California, United States of America

☉ These authors contributed equally to this work.
* bsmarr@ucsd.edu

## Abstract

Type 2 Diabetes causes dysregulation of blood glucose, which leads to long-term, multi-tissue damage. Continuous glucose monitoring devices are commercially available and used to track glucose at high temporal resolution so that individuals can make informed decisions about their metabolic health. Algorithms processing these continuous data have also been developed that can predict glycemic excursion in the near future. These data might also support prediction of glycemic stability over longer time horizons. In this work, we leverage longitudinal Dexcom continuous glucose monitoring data to test the hypothesis that additional information about glycemic stability comes from chronobiologically-informed features. We develop a computationally efficient multi-timescale complexity index, and find that inclusion of time-of-day complexity features increases the performance of an out-of-the-box XGBoost model in predicting the change in glucose across days. These findings support the use of chronobiologically-inspired and explainable features to improve glucose prediction algorithms with relatively long time-horizons.

## Summary

Diabetes mellitus (DM), one of the most common conditions in the world, is a chronic metabolic disease that causes high blood sugar levels. Elevated blood sugar levels can lead to secondary conditions such as heart disease, high blood pressure, and kidney disease. It is therefore important to monitor sugar levels in the blood in order to allow individuals to decide how to best control their diabetes. Noninvasive continuous glucose monitors (CGMs) allow for the monitoring of blood sugar every few minutes instead of the historical self-administered "finger prick" technique. Estimates from CGMs have often be used

**Data availability statement:** The data which supports the findings of this study were made available by Dexcom Inc. Certain restrictions apply to the use and disclosure of the data, which was used under license for the current study. Deidentified, processed data for the machine learning portion of the work is available in the University of California, San Diego Library's Open Data Repository (https://doi.org/10.6075/J0BR8SK9).39 Any requests for the original CGM data should be submitted to Dexcom's Legal Department (privacy@dexcom.com).

**Funding:** The author(s) received no specific funding for this work.

**Competing interests:** The authors have declared that no competing interests exist.

to predict when someone's blood sugar levels may get too high or too low in the nearby future – often within an hour; however, longer-term dysregulation can reflect an individual's overall blood sugar stability. In this study we instead use CGM estimates to predict blood sugar dysregulation on the scale of days, instead of the nearby future, by incorporating information related to the body's ability to self-regulate blood sugar levels across time. We then use machine learning to show that this additional information is better at predicting longer-term dysregulation than typical methods in statistics.

## Introduction

Diabetes mellitus (DM) is a chronic metabolic disease characterized by dysregulated glucose concentration in the blood and is one of the most prevalent medical conditions in the world [1,2]. The most common form of the condition is Type 2 Diabetes (T2D) [3], which occurs when cells and tissues in the body either become resistant to - or do not produce enough - insulin. Insulin critically provides these cells and tissues with the signal to begin absorbing glucose, thus removing it from the blood [4]. Over time, dysregulated blood glucose levels can cause damage to the eyes, kidneys, heart, blood vessels, and nerves [5–7]. Therefore, it is critical to provide people with diabetes with the tools to adequately monitor and control their blood glucose levels so that they are at lower risk for developing the deleterious effects of the condition. One such monitoring tool is a Continuous Glucose Monitor (CGM). These devices are placed on the skin and, using microneedles, estimate a patient's blood glucose concentration by sampling the glucose in the interstitial space between cells [8]. They have become an attractive alternative to older methods of blood-glucose estimation by sampling at high temporal resolution to help avoid hypoglycemic (below 70 mg/dL) and hyperglycemic (above 180 mg/dL) excursions [9,10]. These short-term predictions allow people with diabetes to quickly address their current glucose levels by consuming food or self-administering insulin. While there has been much interest in determining the short-term forecasting of a glycemic excursion [11–15], there are opportunities to investigate how longer-term structures can predict longer-term glycemic dysregulation. Glucose regulation is not driven by a random process. Behavioral patterns [16–19], as well as explained physiological variability [20–23] (e.g., circadian/ultradian biological regulation), contain information about the long-term dynamics of glucose time-series data [24]. Such structures include 1) the variation of glucose levels across multiple days while accounting for the time of day that a sample was taken [25] and 2) the information contained within CGM data over a window of time [26,27]. In this context, information is interpreted as the underlying order or non-random substructures of CGM data. These structures can be used to infer not necessarily a short-term risk of hyperglycemic excursions, but rather a broader risk of an individual's probability to be more dysregulated the next day. While short-term predictions allow individuals to modify their behaviors immediately, long-term predictions give insight into both behavioral and lifestyle changes that could reduce the risk of dysregulation in the first place [28]. To investigate the relationships between these long-term structures and long-term glycemic regulation, longitudinal CGM data is required from a heterogeneous population. In this paper we leverage Dexcom CGM data gathered from 8,000 T2D subjects across 2021 and 2022 to identify long-term structural features that are statistically separable from each other and whose calculation scales efficiently in big time-series datasets. We go on to show that adding these features in an explainable out-of-the-box XGBoost model improves the performance of predicting the severity of glycemic dysregulation compared to using time-series statistics alone. Furthermore, the separation of specific structural features by time of day supports

the hypothesis that biological phases and rhythms are important sources of information in time-series classification problems.

## Results

### Variability of glucose concentrations at the same time of day (ToD) across multiple days reveals an increased risk of glycemic dysregulation

For each subject, we analyzed two weeks of daily CGM records aligned by the clock time at which the samples were taken (Fig 1A). We calculated the within-individual standard deviation separately for each time step (5 min samples) to generate time of day standard deviation

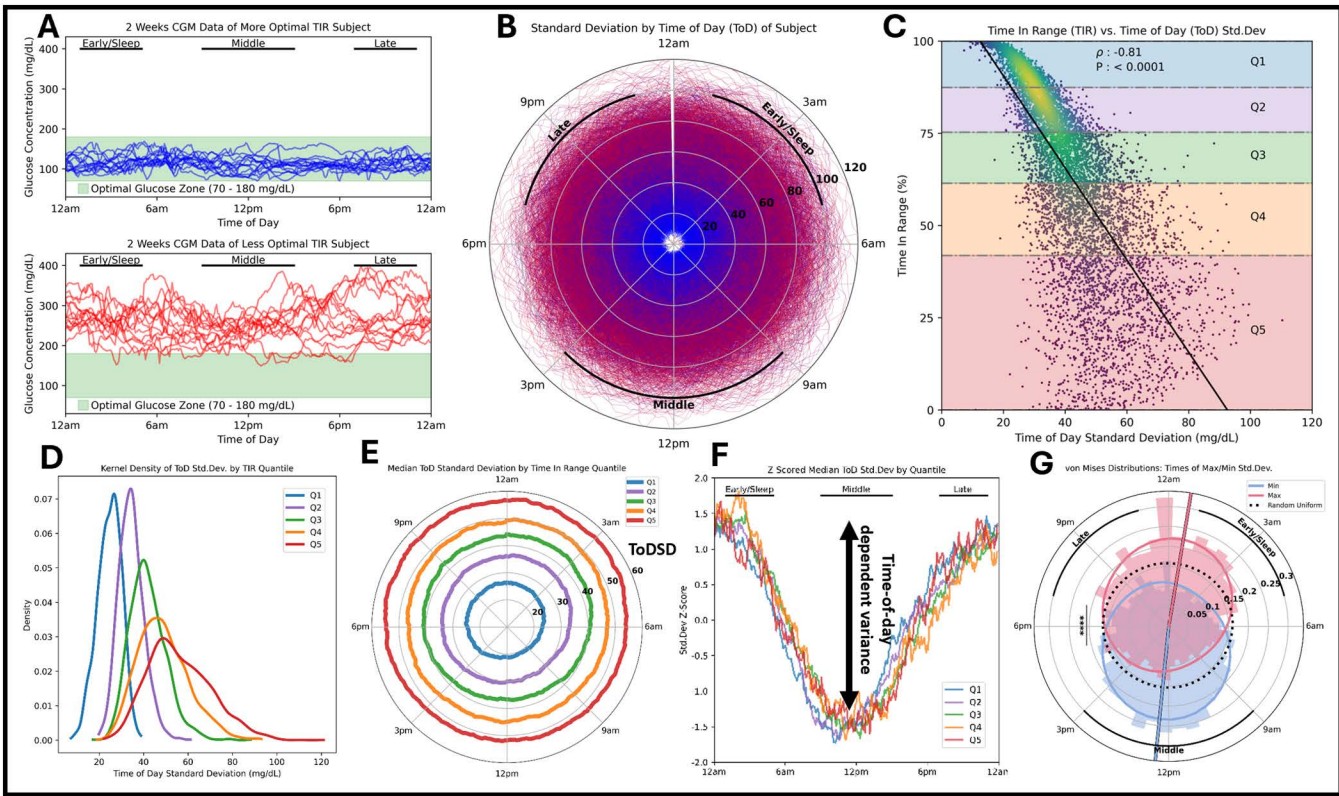

**Fig 1. A. Top – 14 days of continuous glucose monitor (CGM) data for a patient that spends all of their time within an optimal glucose range.** Bottom – 14 days of CGM data for a patient that spends very little time within an optimal glucose range (TIR). **B**. Polar plot of subject CGM standard deviation accounting for time of day across all subjects for two weeks. Two weeks of non-missing CGM data were aligned by time of day (ToD) (12 am to 11:55 pm; 5-minute sampling intervals) for each individual. The standard deviation of the 14 CGM samples at each 5 min time-point was taken and was then projected onto a polar plot to show continuity across 24h. The radius of each polar curve is the standard deviation by person by time point, and the angle around the circle is the time of day. Lines on the polar plot are colored by the average TIR over the two-week period: blue indicates higher TIR and red indicates lower TIR. **C**. Scatterplot of the average TIR over the same two-week period (y-axis) against the average value from each line on the polar plot (x-axis) for all subjects. Points are colored using a kernel density estimate of the axis values' joint probabilities: more yellow indicates a higher density of points, and more purple indicates a lower density of points. Dashed horizontal lines separate the different quantiles of the cohort based on the distribution of TIR values (0th Percentile: 0% TIR; 20th Percentile: 42% TIR; 40th Percentile: 62% TIR; 60th Percentile: 75% TIR; 80th Percentile: 89% TIR; 100th Percentile: 100% TIR). The linear fit of the scatterplot data is visualized as a solid black line. A Spearman correlation was calculated to identify the existence of a monotonic rank-ordering relationship between time-of-day standard deviation and TIR. **D**. Kernel density estimates of the time-of-day standard deviation for each TIR quantile, with the color matching the rectangle shading used in Fig 1C: Quantile 1 has the highest TIR; Quantile 5 has the lowest TIR. **E**. Polar plot of subject CGM standard deviation accounting for time of day across all subjects for two weeks and binned by TIR quantiles identified in Fig 1C. **F**. Unwrapping of the TIR quantile ToD curves onto a time series plot to better visualize the variation of z-scored ToD standard deviation as a function of the time of day. **G**. von Mises distributions of the time of day at which the maximum (pink) and minimum (blue) ToD standard deviation occurs for each 2-week window. A uniform Von-Mises distribution is shown as a black dotted line for clarity.

(ToDSD). ToDSD is then the standard deviation at each time-of-day sample based on the time-matched prior 2 weeks. There is a negative correlation between ToDSD and time in range (TIR) – the percentage of samples that are in ideal ranges (70-180 mg/dL; see Methods for calculation). Individuals with low TIR appeared to have greater variance both for time-matched samples as well as for samples collected within the same day. We observed a potential correlation between ToDSD and the TIR for each patient's 2 weeks of data (Fig 1B). Coloring by TIR, there is a significant gradient from high TIR (blue) to low TIR (red) as ToDSD increased (Spearman rank order test: $\rho$ = -0.81, p < 0.0001) (Fig 1C). The Spearman correlation was chosen since the residuals of TIR are heteroskedastic across 2-week windows when performing a linear fit of the data from ToDSD (Breusch-Pagan test for heteroskedasticity, p < 0.001). The kernel density estimates of ToDSD in the different TIR quantiles can be found in Fig 1D.

We averaged within each 5-min segment of each day across TIR quantiles to elucidate if phases of the day influenced the level of the expected standard deviation of glucose values across multiple days, and if those effects were unique to each quantile (Fig 1E). If there was no influence by the phase of the day, each group's polar plot would have a constant radius around the origin – or a relatively horizontal line on a time series plot. We confirmed Z-scored ToDSD had a non-uniform distribution by time via von Mises Criterion (p < 0.0001). We visually confirmed that all TIR quantiles had a constant radius around the origin when Z-scored (Fig 1F). To identify if this effect was persistent in aggregate, the times of the maximum and minimum ToDSD over each 2-week period were identified and were aggregated into 30-minute bins for the reconstruction of two von Mises distributions (Fig 1G). The von Mises distributions of the times of maximum and minimum ToDSD had circular means of 12:30am and 12:20pm, respectively, and were both kernelized with a peakedness ($\kappa$) of 2 hours. The same test was used between these two distributions, and it was confirmed that the distributions have different means, where the mean of the daily ToDSD minima is around noon and the mean of the daily ToDSD maxima is around midnight (p < 0.0001).

## Multiscale complexity from longitudinal data identifies increased shorter-term variability in individuals with higher TIR

Across-day variance encodes more gross differences in longitudinal CGM time-series; however, there may be information in the shorter-term point-to-point differences. We wanted to determine if there were differences in the average magnitude of short-term CGM fluctuations that may separate individuals based on their observed value of daily TIR estimates. For each subject, we sampled and Z-scored 30 days of their CGM data. Using contiguous, non-overlapping windows of increasing length $L$, we calculated the average value of each window of $L$ glucose concentrations to generate a coarsely grained CGM time series. We then calculated the complexity index (CI – see Methods) of each reduced time-series to create a plot of CI against coarsing scale, or averaging window size, $L$ for each patient month (Fig 2A) which we termed the multiscale complexity index (mCI). We chose the separation of "small" window sizes from "large" window sizes heuristically as the window size at which the largest proportion of low TIR (red) curves reached their maximum CI – in this case that size being 24 samples (2 hours of 5-minute samples). Window sizes less than or equal to 24 samples were assigned as "small" sizes, and any size greater were assigned "large". The algorithm is performed on the standardized (Z-scored) sequence of data at each coursing scale (as recommended by the original authors (see Methods)) of the Complexity Index since their original aim of the metric was to identify if different sequences had greater short term (point-to-point) fluctuations relative to the global variance of the data. The greater Complexity Index values in

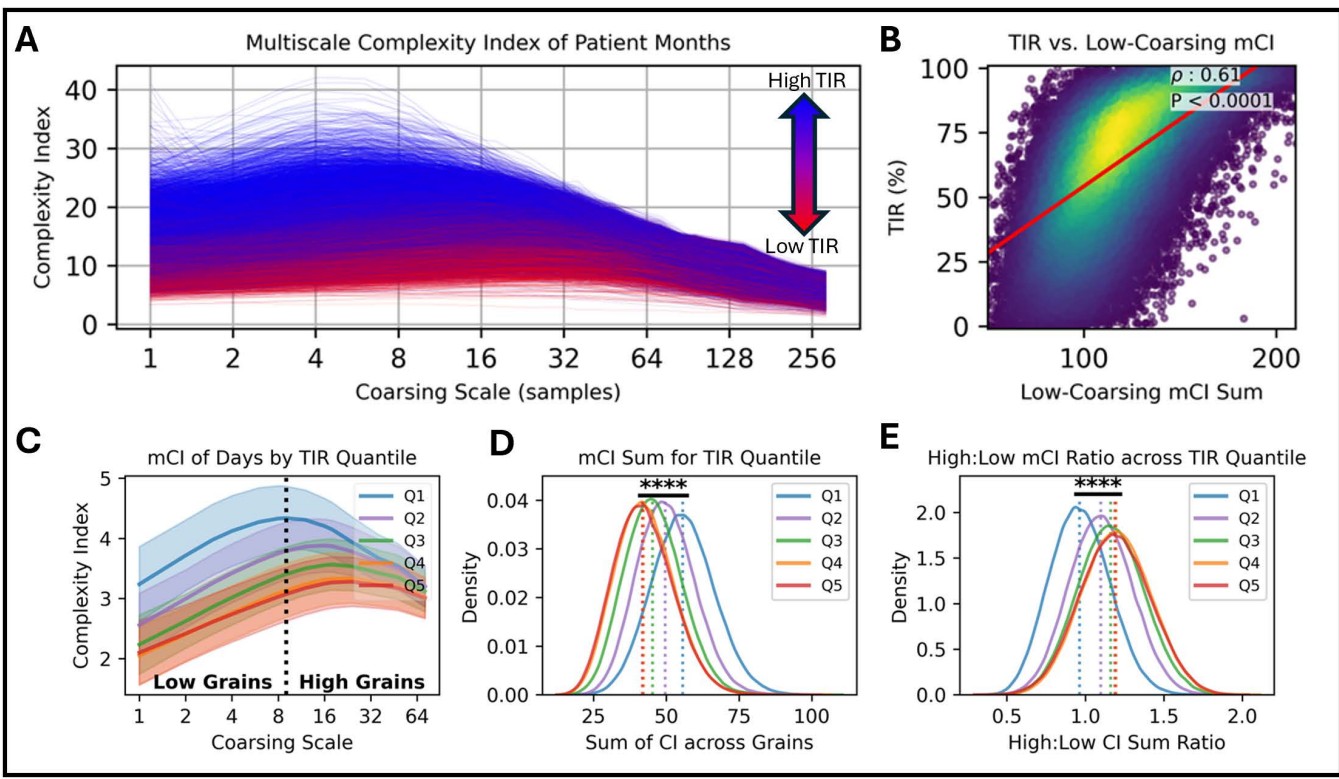

**Fig 2. A. mCI curves derived from 30 days of CGM data across all individuals.** Coarsing scales (averaging window size) range from 1 (5-min samples) to 288 (1-day averages). Multiscale complexity index (mCI) and coarsing are described in Methods. Curves are colored by the average time in range (TIR) over the 30-day period: blue indicates higher TIR and red indicates lower TIR. B. The average TIR over the 30-day period plotted against the sum of the complexity indices between a coarsing of 1 (5 min) thru 12 (1 hour). Points are colored using a kernel density estimate of the axis values' joint probabilities: more yellow indicates higher density of points, more purple indicates lower density of points. Spearman's rank-order correlation was calculated to identify the existence of a monotonic rank ordering relationship between sum of complexity indices at smaller window sizes and TIR. An approximated linear trend is indicated by a red line. C. mCI curves binned by TIR quantile. The vertical dotted line denotes the scale separation between low and high grains. D. Kernel density estimation of the sum of the CI across all scales for each mCI curve in each quantile bin E. Ratio of the sum of high grain CI to low grain CI for each mCI curve in each quantile bin.

the CGM data for individuals with greater time in range reflects that short term fluctuations describe a greater proportion of global variance compared to individuals with less time in range (examples of CI from a High Time in Range and Low Time in Range individual can be found in S1 Fig). Since the "small" averaging window sizes appeared to have greater variance than the "large" window sizes, they were chosen to be correlated against the TIR for each subject. There is a significant positive correlation between the sum of small window CI and the average TIR for each day over the 30-day period (Spearman rank order test: $\rho$ = 0.64, P < 0.0001) (Fig 2B).

To test if similar separability was identifiable over shorter time windows, we ran the mCI analysis on each day (288 5-minute samples) of each subject while using a maximum window length L of 72 samples (6 hours). We binned all curves into their TIR quantiles determined by the TIR for that day (Fig 2C). First, we calculated the total sum of the CIs at each scale to determine the magnitude difference between quantiles (Fig 2D). All medians of the CI sums were significantly different from each other (Kruskal-Wallis H test: P < 0.0001; Bonferroni corrected *post hoc* Dunn's test: P < 0.0001 between all distribution medians). We chose the window scale of 9 samples (45 minutes) as the separation of "small" and "large" window sizes for an even comparison of

the area under the small/large window portion of the curves for each quantile (Fig 2E). The ratio of large window sums to small window sums was used as a comparison of longer-term fluctuations to shorter-term fluctuations that was independent of CI magnitude. All medians of the CI ratios were significantly different from each other (Kruskal-Wallis H test: $P < 0.0001$; Bonferroni corrected *post hoc* Dunn's test: $P < 0.0001$ between all distribution medians).

### Accounting for days that have nearly perfect TIR or nearly no TIR, mCI decreases with age

To elucidate if the variance within the quantile-binned multiscale complexity curves was caused by random effects or possible subject phenotypes we controlled for the level of TIR and binned into High TIR (HTIR) and Low TIR (LTIR) groups (see Methods for selection criteria). While controlling for these TIR extrema, we further subdivided the groups of CI curves by age tranches of 10 years (Fig 3A). mCI curves contained within the HTIR group are indicated by a blue-to-green gradient, while curves contained within the LTIR group are indicated by a red-to-purple gradient. There are visible trends that indicate decreasing low coursing complexity sum as a function of increasing age in both the high (Spearman rank order test: $\rho = -0.95$, $P < 0.0001$) and low (Spearman rank order test: $\rho = -0.65$, $P < 0.0001$) TIR groups (Fig 3B).

### Extracted features for across-day variance and complexity improve performance in an out-of-the-box machine learning model and contribute linearly orthogonal information to the prediction

To test whether the statistically separable features add predictive information, we next deployed them into an out-of-the-box XGBoost model (see Methods) to determine if they

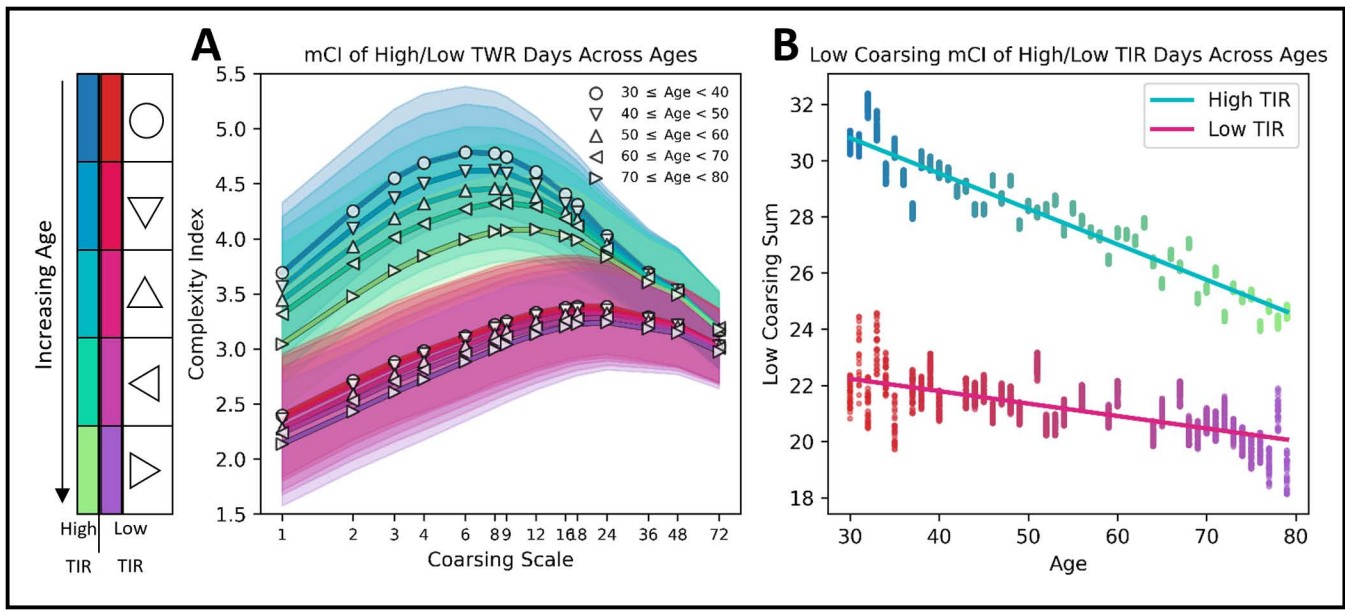

**Fig 3. A. mCIs for 30 days of CGM data across different graining window sizes.** A very high time in range (TIR) group (blues) and a very low TIR group (pinks). Groups were separated further into tranches of age (10-year bins). Lines for each age group are annotated both by color and marker style. B. The sum of complexity indices for a coarsing window size less than or equal to 12 (1 hour) was plotted against the age of each individual. Upper blue points represent the very high TIR group; lower purple points represent the very low TIR group. Scatter points are colored by age, matching the color gradient in the left figure. Linear regression was used to fit trend lines to the high (blue) and low (red) coursing sums across age.

could improve model performance compared to using standard statistical features alone. Model 1 has statistical features, whereas Model 2 also includes the across-day variance and complexity features. The inclusion of the complexity and across-day variance features improved one-vs-rest model performance as determined by the area under the receiver-operator curve (AUROC) for all three labels (0: lesser area above range; 1: similar area above range; 2: greater area above range) compared to an equivalent model using statistical features alone (Fig 4A–4C) (Calibration curves for the models can be found in S2 Fig). Since the XGBoost model is capable of quantifying feature importance, we compared the information gain of features between the models to identify the contribution of the complexity and

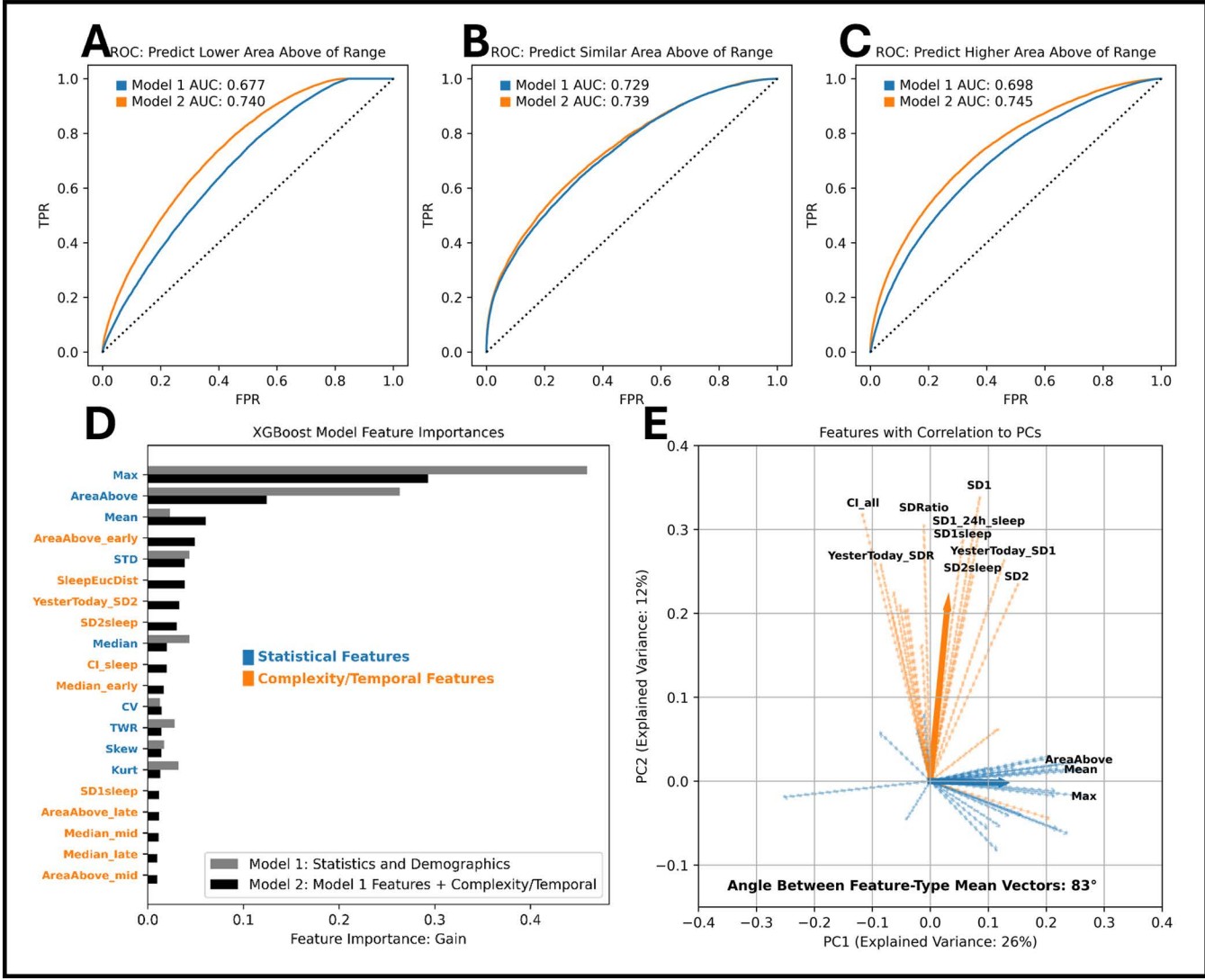

**Fig 4. A. Receiver-operator curve (ROC) for the prediction of class 0 (area above range is lower the next day). B**. ROC for the prediction of class 1 (area above range is similar the next day). **C**. ROC for the prediction of class 2 (area above range is higher the next day). **D**. Horizontal bar graph of the feature importance values for each feature in the model. Feature importance is based on "Gain" – the increase in model accuracy when a feature is used to perform a split on a parent node. The "types" of features are noted by font color, and model types are separated by the utilization of certain features. **E**. Plotting of the correlation of all features to the first two principal components determined from PCA. Only features that are greater than the average correlation are annotated for clarity. Feature "types" are annotated by color.

across-day variance features. The information gain refers to the increase in model accuracy when a feature is used to perform a split on a parent decision boundary/node. The top 20 most important features were plotted together to show their relative importance across models (Fig 4D. If the novel features added predictive power to the model, then importance would likely decrease from the statistical features and increase in the complexity/temporal features. While the statistical aggregation features contributed the most gain to both models, 5 features generated from complexity or across-day variance were in the top 20 list when creating Model 2: SleepEucDist, YesterToday_SD2, SD2sleep, CI_sleep, SD1_sleep (a list of all features and descriptions can be found in S1 Table). However, it remained unclear whether the novel features were contributing unique information to the model and were not, in fact, colinear with the original features. We performed principal components analysis (PCA) to determine if the variance explained by the complexity and across-day variance features were largely orthogonal to the variance explained by the statistical aggregation features (Fig 4E). The angle between the mean vectors of correlation for the original (blue) and novel (orange) features along each of the first two principal components (PCs) was calculated to be 83°, indicating that the complexity and across-day variance features contributed unique, orthogonal information about the target variable compared to the statistical features (Table 1).

## Discussion

Here we demonstrate that chronobiologically-informed features provide unique information about an individual's level of glycemic regulation across days. We first confirmed that ToDSD is a powerful identifier of within-individual glycemic variation and how it relates to TIR. We also identified that minimum and maximum ToDSD occur at around noon and midnight, respectively. Glucose utilization has been observed to be lower during sleep in people with diabetes, leading to plateaued, elevated blood-glucose concentrations [29]. Alternatively, during wakefulness, glucose utilization is higher and leads to a decay of blood-glucose concentration toward a resting value [30,31]. Since ToDSD estimates glucose variance across multiple days within an individual, we believe the midnight maximum for the entire sample may reflect lack of glucose regulation during sleep. The intuition for this is that since glucose concentration prior to sleep onset is affected by the carbohydrate content and timing of a meal, and since that concentration will plateau during sleep, the greater glucose variance is incurred by variance in day-to-day eating behaviors of an individual. The minimum ToDSD around noon may be reflective of an increase in glucose utilization and regulation, which will assist in driving concentrations down by being proportional to concentration itself. To capture different timescales of variance as functions of time-of-day, we developed a novel modification of the complexity index [32] using the information theoretic concept of course graining [33], which

**Table 1. Table of model performances for Models 1 and 2 including (from left to right) area under the receiver operating characteristic curve (AUROC), precision, recall, F1-score, and overall data support.**

|             | AUROC | | Precision | | Recall | | F1-Score | | Support |
|-------------|-------|------|-----------|------|--------|------|----------|------|---------|
| Model       | 1     | 2    | 1         | 2    | 1      | 2    | 1        | 2    |         |
| Lower AAR   | 0.67  | **0.75** | 0.5   | **0.56** | 0.68 | **0.72** | 0.58 | **0.63** | 127873 |
| Similar AAR | 0.72  | **0.73** | **0.65** | 0.59 | 0.17 | **0.23** | 0.28 | **0.33** | 63887 |
| Higher AAR  | 0.70  | **0.76** | 0.57  | **0.61** | 0.57 | **0.62** | 0.57 | **0.61** | 128825 |
| Accuracy    |       |      |           |      |        |      | 0.53     | **0.58** | 320585 |
| Macro Avg.  | 0.70  | **0.75** | 0.57  | **0.59** | 0.47 | **0.53** | 0.47 | **0.53** | 320585 |
| Weighted Avg. | 0.69 | **0.75** | 0.56 | **0.59** | 0.53 | **0.58** | 0.51 | **0.56** | 320585 |

we label multiscale complexity index, or mCI. We found that the mCI of CGM data is positively correlated with TIR and further decreases in proportion to subject age. We also observe from the XGBoost's feature information gain that the mCI during a subject's sleep provides more separability than using the mCI of the entire day. This is consistent with our hypothesis that the effects of meal timing and circadian rhythms/sleep could be revealed in CGM data, and that chronobiological dynamics would thereby affect the structure of CGM data. Having confirmed that these chronobiologically-informed features are aligned with our expectations of how they may be related to glycemic regulation, we trained an XGBoost model on statistical features alone and compared it to an equivalent model that further included the temporal and mCI features. The XGBoost model was chosen due to its propensity to outperform other classification models and ability to handle potentially nonlinear relationships. Furthermore, since we were primarily concerned with the improvement in model performance with the addition of chronobiologically-informed features, we did not tune the model since the goal was to determine the additional feature information versus the extraction of maximal performance from the model. The top 3 features for the model appear to reflect quality of the "naïve" guess in that the AAR magnitude the day prior for an individual is a strong predictor of AAR magnitude the following day. The complexity/temporal features then add additional information that improve upon this naïve assumption. More specifically, with the exception of the YesterToday_SD2, the top 6 complexity/temporal features are extracted from CGM data sampled during the estimated sleep window. If additional data such as true sleeping windows as well as caloric intake and timing were available, it would be possible to investigate if the importance of CGM during sleep windows is associated with underlying uncontrolled physiological variance or differences in eating habits between individuals with T2D. We found improvement in the model's AUROC when adding these features, and additionally confirmed that these added features were predominantly orthogonal vectors after using PCA, indicating the information was independent from the typical statistical features. While this work would have been strengthened by comparing our model's performance with models developed by other groups that use similar features and labels, to our knowledge no other models exist for this specific task. For this reason, we focused less on optimal model architecture and more on performance improvement with novel feature engineering that may incorporate some biological and/or behavioral effects. Our hope is that other groups may be able to adopt these feature engineering approaches and tune the outcome variable of interest to improve upon biologically-informed featurization for ML prediction tasks.

Glucose metabolism is coordinated by the circadian system, and perturbations that can desynchronize or misalign these rhythms potentially lead to adverse health outcomes [18,21,34]. We did not have labels for the times at which subjects ate or direct confirmation of the times at which subjects were sleeping, which limited our ability to identify effects in glycemic regulation caused by fasting, post-prandial spikes, and sleep. Time-of-day changes may be largely impacted by behaviors, but glycemic complexity can be interpreted as being an emergent structure of multiple interacting components that regulate glucose metabolism [27,35]. We chose to focus on hyperglycemia as it is typically considered to be more related to mismanaged/mistreated blood-glucose concentration, the primary clinical endpoint of this work, whereas hypoglycemia is usually a side effect of blood-sugar-lowering medications [36]. Since the data provided did not include factors such as ethnic group and socioeconomic status of the participants, which are known to affect outcomes of people with diabetes [2,37], we hope that future analyses would attempt to better understand the significant effects found in this work when accounting for such demographic details. Furthermore, since the usage of different CGM devices are not geographically or socioeconomically homogenous due to differences in local availability, cost, insurance coverage, *etc.*, there may be representative bias

in the underlying population using Dexcom G6 that may not be proportionally similar to populations that use a different device. However, the features extracted to analyze the continuous data were inspired by prior work in controlled settings, and may hopefully be more generalizable to different devices. Several clinical studies have identified the statistical differences in multi-scale sample entropy measurements (an information-theoretic estimation of signal complexity) between groups with no diabetes and groups with diabetes [26,27,35]. These calculations may be suitable for low sample sizes over a shorter period, but they are not scalable when calculating from larger T2D populations over a much longer time window. To overcome this limitation, we developed a new multi-scale version of the complexity index (mCI) to allow for faster data preprocessing and model development times. A higher mCI generated from narrow graining windows likely indicates that an individual is more responsive to acute perturbations in their blood-glucose levels. By contrast, individuals with substantially higher mCI from graining windows around 1-2 h likely reflect more stable BG, which changes in response to meals. We further validated this new mCI construct by finding that it reflects age-related decreases in complexity, which can be inferred to be related to "critical slowing down" of physiological responses to perturbations in aging populations [38]. This age-related decrease was strongly apparent even when controlling for days where nearly all CGM estimates fall within range. Finally, our mCI proved to be a robust metric that provides unique information about the relationship between patient demographics and glucose metabolism stability.

Long-horizon glucose predictions are accompanied by substantial stochastic baggage due to a subjects' behavior changing features over time. Over longer periods of time, it is unknown when a subject will eat, how much they will eat, what the glycemic index of that food is, along with a plethora of other factors that cannot be accounted for while looking so far into the future. An alternative to predicting uncertain futures is providing decision support services. In such a model, the algorithm is used to predict the most likely trends given recent past behaviors. This approach can support behavior modification or reinforcement while also reducing false precision of longer-term predictions; however, identifying the ways in which a "closed loop" version of this model that changes how individuals modify their behaviors would require a prospective study. The features that are extracted in this work are primarily meant to act as potential features should such a prospective study be performed. T2D individuals cannot directly control their circadian system to regulate their glucose more effectively, but they can control the behaviors that may misalign their rhythms in the first place. Allowing these features to be conserved when being used for longer-horizon glycemic dysregulation prediction provides CGM users with information about which behaviors or disruptions had the highest impact on their next-day prediction. The extraction and implementation of features that increase model performance and interpretability are critical to giving individuals with T2D the information and agency necessary to affect their future trends. Future work can identify best practices when using such decision support algorithms in prospective field studies. The overall goal of the

In summary, we present an interpretable, longer-time horizon, glycemic dysregulation risk model, and demonstrate that model performance is improved by the inclusion of time-of-day-dependent features and complexity features. These engineered features also increase prediction explicability. Our findings support the use and further exploration of chronobiologically-informed features to increase model performance for glycemic regulation or prediction algorithms.

## Methods

### Ethics statement

All data collection and data privacy pertaining to human participants adheres to the guidelines set out by the University of California, San Diego Office of IRB Administration. The office has

waived the need for ethical approval (IRB: #811908) and informed consent for this study given that it falls under exempt status: Research involving the collection or study of existing data, documents, records, pathological specimens, or diagnostic specimens, if these sources are publicly available or if the information is recorded by the investigator in such a manner that subjects cannot be identified, directly or through identifiers linked to the subjects (45 CFR 46 Subpart A, Paragraph 4).

### Data description

Dexcom provided a subset of CGM data sampled at 5-min intervals from T2D customers that were utilizing the company's G6 device. The cohort comprising the data was heterogeneous with respect to reported gender, age, and treatment type (Table 2).

The data provided by Dexcom does not contain any identifiers and is in compliance with HIPAA regulations and other relevant privacy standards.

### Calculation of time in range and area out of range

Time in range (TIR) was calculated as the percentage of samples within the optimal glucose concentration range (70 – 180 mg/dL). For a single day there were 288 5-min samples; therefore, for a day where all samples are in range, a subject would have 100% of their samples (288/288) of their data within range.

Area Above Range (AAR) was calculated as the area under the curve (estimated using the trapezoid rule) for the glucose concentration estimates that are above the 180 mg/dL upper optimal limit. On a given day, a subject can have all CGM data above 180 mg/dL, but the AAR may be different across days that have the same TIR.

### Variability of glucose concentrations at the same time of day (ToD) across multiple days reveals an increased risk of glycemic dysregulation

Two weeks of non-missing CGM data were aligned by time of day (12 AM to 11:55 PM; 5-min sampling intervals) for each individual starting from May 9, 2022 through May 22, 2022, as this was a window of time that met two criteria: 1) all individuals had CGM data during this time and 2) no individual had more than an hour of CGM data missing per day. The standard deviation of the 14 CGM samples at each 5 min time-point was taken and was then projected onto a polar plot to show continuity across 24h.

Table 2. Demographic information of Dexcom Type 2 diabetes cohort.

| Demographic | Category | Number of Subjects |
|---|---|---|
| Reported gender | Male | 4000 (50%) |
| | Female | 4000 (50%) |
| Age (years) | 30–39 | 450 (6%) |
| | 40–49 | 1550 (19%) |
| | 50–59 | 2000 (25%) |
| | 60–69 | 2000 (25%) |
| | 70–79 | 1618 (20%) |
| | 80+ | 382 (5%) |
| Treatment type | Fast acting | 714 (9%) |
| | Long acting | 1214 (15%) |
| | Both | 4895 (61%) |
| | Neither | 1178 (15%) |

## Coarse graining

Multiscale coarse graining is the process of producing several new time-series by reducing the length of an original time-series by using an aggregation function over a subset of samples based on a varying window size. For the purposes of this work, window sizes were integers based on the divisors of 288 glucose samples that leave a remainder of 0 to ensure no data-points were unnecessarily dropped. The sample mean was used as the aggregation function over each non-overlapping window of glucose data. More formally, for each subset $s$ of $\tau$ glucose samples in a CGM time-series $G(n)=\{g_1,g_2,...,g_{n-1},g_n\}$ of size $n$, the coarsely grained value of that subset can be calculated as

$$\tilde{g}_s^\tau = \frac{1}{\tau} \sum_{s\tau}^{i=\tau(s-1)+1} g_i, \ 1 \le s \le \frac{N}{\tau} \tag{1}$$

to create a new, coarsely grained time-series $\tilde{G}^\tau(s) = \left\{ \tilde{g}_1, \ \tilde{g}_2,...,\tilde{g}_{\frac{N}{\tau}-1}, \tilde{g}_{\frac{N}{\tau}} \right\}$.

## Complexity Index

The calculation of the Complexity Index (CI) was derived from work established by Batista et al. [29] In summary, the metric is used to determine the "effective length" of a time-series if it were to be extended until flat. More formally, for a discrete, uniformly sampled time-series $X(n)=\{x_1,x_2,...,x_{N-1},x_N\}$, the CI can be calculated as:

$$CI_{X(n)} = \sqrt{\sum_{N-1}^{n=1} \left( x_n - x_{n+1} \right)^2} \tag{2}$$

To ensure that the comparison of CI is consistent across time-series with substantially differing variances, all CGM data is Z-scored and the CI is normalized by the quantity of samples used in its calculation (examples of CI from a High Time in Range and Low Time in Range individual can be found in S1 Fig).

## Selection of high and low TIR groups

The High and Low TIR (HTIR and LTIR) groups were selected as the group of days in which the TIR over the course of the entire day is greater than or equal to 20 hr or less than or equal to 4 hr, respectively.

## Multiscale complexity from longitudinal data identifies increased shorter-term variability in individuals with higher TIR

The 30 days of CGM data were selected based on similar criteria as the two weeks used in the ToDSD analysis section, which ended up being May 9, 2022 through June 7, 2022.

## XGBoost model

The out-of-the-box model was generated using the *xgboost* Python library (number of estimators: 100; maximum depth: 4 layers). The target variable was if a subject would increase, decrease, or stay relatively similar to (±20%) their area above optimal glucose range on a given day $D$ relative to their area above range on day $D-1$. All features extracted from the time series were obtained using the CGM data between midnight on day $D-2$ through 5 AM on day $D$, thus using a total of 53 hr of data (a full 44-feature list can be found in S1 Table); the implementation premise being that a subject could wake up to information about their risk of increased area out of range if notified through a mobile device. An 80:20 train-test split was

performed on 320,585 samples. The same random state parameter was used in the data splitting and model training/testing to ensure the only difference between samples was the features used. Both models were fit with 5-fold cross validation (S2 Table and S3 Table for Model 1 and Model 2, respectively).

## Software and statistics packages

All data analysis was performed in Python version 3.11.9. The following libraries were used for data analysis and statistical testing: *pandas* (v2.2.2) for dataframe manipulation; *numpy* (v1.26.4) for array manipulation, data transformations, and numerical calculations (including the multiscale complexity index); *scipy.stats* (v1.13.1) and *statsmodels* (v0.14.2) for statistical testing; *matplotlib* (v3.8.4) for plotting and data visualization; *xgboost* (v2.1.2) for classification model development; *sklearn* (v1.5.0) for calculation of model performance metrics.

## Supporting information

**S1 Fig. Example of low and high time in range CGM data and their respective multiscale complexity index curves.**
(TIF)

**S2 Fig. Left. Calibration curves for 3-label classification in Model 1 (statistical features only). Right. Calibration curves for 3-label classification in Model 2 (Model 1 features + temporal and complexity features).**
(TIF)

**S1 Table. List of features used in the XGBoost Model.**
(XLSX)

**S2 Table. Model 1 performance for each fold of training and testing.**
(XLSX)

**S3 Table. Model 2 performance for each fold of training and testing.**
(XLSX)

## Author contributions

**Conceptualization:** Jamison H. Burks, Benjamin Smarr.

**Data curation:** Jamison H. Burks, Leslie Joe, Karina Kanjaria, Carlos Monsivais, Kate O'laughlin.

**Formal analysis:** Jamison H. Burks.

**Investigation:** Jamison H. Burks.

**Methodology:** Jamison H. Burks.

**Project administration:** Jamison H. Burks, Benjamin Smarr.

**Resources:** Benjamin Smarr.

**Supervision:** Benjamin Smarr.

**Validation:** Jamison H. Burks.

**Visualization:** Jamison H. Burks.

**Writing – original draft:** Jamison H. Burks, Benjamin Smarr.

**Writing – review & editing:** Jamison H. Burks, Leslie Joe, Karina Kanjaria, Carlos Monsivais, Kate O'laughlin, Benjamin Smarr.

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
