## [Decision Letter · Decision Letter 0]

5 Sep 2024

PDIG-D-24-00260

Chronobiologically-Informed Features from CGM Data Provide Unique Information for XGBoost Prediction of Longer-Term Glycemic Dysregulation in 8,000 Individuals with Type-2 Diabetes

PLOS Digital Health

Dear Dr. Smarr,

Thank you for submitting your manuscript to PLOS Digital Health. After careful consideration, we feel that it has merit but does not fully meet PLOS Digital Health's publication criteria as it currently stands. Therefore, we invite you to submit a revised version of the manuscript that addresses the points raised during the review process.

Please submit your revised manuscript within 60 days Nov 04 2024 11:59PM. If you will need more time than this to complete your revisions, please reply to this message or contact the journal office at digitalhealth@plos.org. Please include the following items when submitting your revised manuscript:

We look forward to receiving your revised manuscript.

Kind regards,

Hisham Essam Hasan, M.Sc.

Guest Editor

PLOS Digital Health

Journal Requirements:

1. Please upload your main article file as a .doc, .docx or .rtf file.

Additional Editor Comments (if provided):

Please provide separate figure files in .tif or .eps format only and remove any figures embedded in your manuscript file. Please also ensure that all files are under our size limit of 10MB. For more information about figure files please see our guidelines:

The manuscript does not comply with PLOS's data availability policy. You must make all underlying data fully available without restriction, either as part of the manuscript or deposited in a public repository. Please address this issue promptly.

The reviewer was uncertain about the rigor of the statistical analysis. Ensure that all statistical methods are clearly described and validated, and consider including a detailed explanation of your analysis approach.

The manuscript needs improvements in language clarity and presentation. There are typographical errors, unclear explanations, and a need for better structure in presenting methodology and results.

The current validation approach using a train-test split is a good starting point but insufficient. I recommend incorporating cross-validation methods to enhance the robustness of your model's performance. Additionally, external validation using independent datasets would strengthen the generalizability of your results.

While the study emphasizes chronobiologically-informed features, a deeper exploration into their interpretability and clinical relevance is needed. Discuss how these features can translate into actionable insights for patients and clinicians, providing examples or case studies to illustrate their practical implications.

Include a more detailed discussion on the selection process for the chronobiological features and their relative importance in the model. A comprehensive analysis of which features contribute most to the predictive power of the XGBoost model would be beneficial.

Acknowledge and address potential biases related to the CGM data provided by Dexcom. Discuss any inherent biases or limitations of the dataset and their potential impact on the generalizability of your findings.

Provide a comparative analysis with other state-of-the-art methods or models for predicting long-term glycemic dysregulation. This will help contextualize the performance gains achieved with your approach.

Figure 1:

Clarify x-axis labels in panel D and provide a legend explaining quantile ranges.

Include additional annotations in panels E and F.

For panel G, provide more details on the von Mises distribution.

Figure 2:

Add grid lines in panel A for better trend visualization.

Include trend lines or additional statistical summaries in panel B.

Provide explanatory text or labels for key trends in panel C.

Include information on statistical significance in panels D and E.

Figure 3:

Add more explicit labels or annotations in the left panel.

Provide detailed statistical summaries or trend lines for gender-specific effects in the right panel.

Figure 4:

Summarize key performance metrics directly on the plots in panels A-C.

Provide a brief discussion of impactful features in panel D and highlight key takeaways or additional annotations in panel E.

Table 1:

Include percentages for gender distribution, age categories, and treatment types to enhance clarity.

Confirm that the data provided by Dexcom is de-identified in compliance with HIPAA regulations and other relevant privacy standards. Ensure that no identifiers are present, either directly or indirectly, that could lead to the identification of individual subjects.

Although the IRB waiver applies, mentioning its reference number is crucial. Please include this reference in your revised manuscript.

Discuss any potential ethical implications of the findings, particularly if the research could influence diabetes management practices or patient outcomes. Address how the findings will be communicated and used to ensure that they do not inadvertently harm or mislead patients.

Reviewers' comments:

Reviewer #1: There are some comments regarding the submitted paper. However, the peer review process would be easier if manuscript lines were numbered to cite a certain line.

Major

• How the authors justify for the ToDSD in the noon be the lowest and on the midnight be the highest. It’s known that the sensitivity for insulin decrease in the night and the glucose peaks at this time. Is there any evidence from the literature about this observation? 

• The authors used certain time frames such as two weeks and one month as a sample of time from each patient. However, the authors didn’t mention how they chose that time for each patient. Is it the same time of the year for all patients or was it sampled randomly or more than one period were taken from each patient?

• In Figure 3, there is no explanation or statistical evidence provided for the differences in mCI after controlling for the gender. The paper is presented as there is already a significant difference.

Minor

• The authors didn’t consider for the epidemiological factors in the study except for the age. These factors could include the ethnic groups and socioeconomic status for the patients included in this study. This should be stated at least as a limitation.

• To facilitate study repeatability, the authors should state how they performed calculations and processes on the data and using which software as they only included the machine learning part.

• The authors mentioned certain observations in the figures without further analysis and discussion. See e.g. the last sentence in the first result paragraph.

• The authors consistently present the results at the beginning of each paragraph, followed by a description of how they were measured. For example, in the statements "There is a negative correlation between ToD Standard Deviation (ToDSD) and time in range" and "The magnitude of ToDSD across multiple days is affected by ToD itself. In aggregate, subjects have a maximum and minimum ToDSD at around midnight and noon, respectively," as well as in other paragraphs. It would enhance clarity for the reader if the methodology or measurement process were discussed first, followed by the results. This approach would reduce potential confusion and make the findings more accessible. I recommend additional academic language editing to improve the overall readability of the paper and correct tenses.

• In Figure 1C, why does the range for average hours of TIR for patients span from 0-100? Was the data normalized?

• Figure 3 consists of two charts, but they are not labeled with distinguishing letters or identifiers.

• There is a spelling error in “glyc1emic” at the paragraph just before the last one in discussion.

Reviewer #2: In this msnuscript, the authors used Dexcom CGM data gathered from 8,000 T2D subjects to identify long-term structural features. They showed that adding these features in an explainable out-of-the-box XGBoost model improves the performance of predicting the severity of glycemic dysregulation compared to using time-series statistics

alone. The idea behind the research is interesting, however, ther are some point that should be considered:

- Why did the authors use only XGBoost model? They should use other ML algorithms and compare the results.

- The parameters such as Accuracy, Precision, Recall, and F1-score should be mentioned.

- In "Discussion", the authors should compare their results with the previously published papers and discuss about the advantages of their model.

-Overall, the manuscript has been written as a report, not a research paper.

Reviewer #3: Burks et al. investigated whether chronobiologically-informed features from continuous glucose monitoring (CGM) data has additional value on top of classical CGM features for prediction of glycemic regulation (longer term=next day). Some of these features and their association with classical CGM measures (TIR) are informative and intuitive (e.g. SD at different time points), while others are to a lesser extent (complexity index, see details below). In general, the study focuses on an interesting topic that is often overlooked, but some aspects could be strengthened, especially from a clinical perspective (there doesn’t seem to be a clinician among co-authors). The dataset is large for a CGM study and the statistical analysis was conducted with care, although the reporting could be more detailed (see details below). Neither data nor code are publicly available. I have the following questions/suggestions to further improve the manuscript:

Major

Chronobiologically-informed features

As the authors described, most other studies used short-term features for prediction, so I really support extending this horizon. However, I don’t find the complexity index and its positive association with TIR clinically intuitive. CI (I suggest to avoid using this abbreviation, because 99% of readers with a medical background will confuse it with confidence interval), is based on the difference of consecutive measurements, so if someone has a completely flat curve in range, then it would be 0, if I understand correctly. At the same time, if someone has huge swings (which is also one way to have higher TIR), then CI would be expected to be higher. So for me it’s not intuitive how high complexity is associated with high TIR. Could the authors include a couple of examples (similar to Fig 1A) to demonstrate a couple of scenarios/profiles? I would like more arguments for looking at age- and sex-related associations (Fig 3), which I don’t find too relevant, but it might be because of the lack of intuition in general about CI.

Choice of outcome/target 

I’d like the authors to justify why this is the relevant outcome to predict. If a person had really good control (e.g. close to 100% TIR), then there is only one way to go from there: towards a higher area above range (AAR). The other way around is also true, with high values for the last two days, one is more likely to improve, which is the well-described phenomenon of regression to the mean. This is also reflected in what the top 3 strongest predictors are. If the authors argue for using this outcome, then please consider doing sensitivity analyses using a different threshold in the definition of the outcome (e.g. 10% change instead of 20%), or argue why 20% was chosen. Please also report the frequency of the three categories of the outcome. 

I was also wondering why there was focus on hyperglycemia and not hypo (below range).

Based on the following sentence: ‘long-term predictions give insight into both behavioral and lifestyle changes that could reduce the risk of dysregulation in the first place’, it seems to me that the authors are interested in using the model to make individual recommendations to individuals. This is often done by using thresholds, e.g. if the probability is above 20% then do the following action (e.g. exercise)… In this case, discrimination is not the best performance metric to report. Calibration (e.g. calibration in the large and calibration slope, see BMJ 2024; 384 :e074819) would be more relevant, because that gives an idea of the magnitude of predicted probabilities vs actual outcomes.

Predictors

I am curious why only two days are used as predictors and why a longer period was not utilized. The interesting results described in Fig 1 are not really utilized in the prediction model as far as I understand.

I suggest the authors include all predictors tested and how they were defined in a supplement. The top 20 are presented, but the names on Fig 4D are not all self-explanatory (e.g. YesterToday_SD2). Please include the definition of all of these.

‘If they provided predictive power then they would appear in the top 20 list’ Just because a predictor appears in the top 20, it doesn’t mean that it is relevant, if all the other tested predictors are irrelevant, so I suggest reformulating this. The authors then mention that 5 complexity features made it into the top 20, but I can only find CI_sleep that explicitly mentions CI. Which four am I missing?

Minor

I suggest using ‘people with diabetes’ instead of DM subjects/diabetic individuals, or participants instead of subjects throughout the manuscript (Diabetes Care 2017;40(12):1790–1799).

Fig 1C the unit of TIR is % I suspect and not hours.

Have the authors considered separating weekends in the analysis and creating weekday- and weekend-specific features? People often have very different behavioural patterns in these two periods.

It is not very reader-friendly to have the Methods section at the end of the manuscript as there are several places in the Results where the authors refer to it. If it’s journal style, then it is a comment rather to the editorial board, otherwise I suggest moving the Methods before the Results.

Please make the figures self-explanatory i.e. introduce all abbreviations again.

Please improve the resolution of some figures (e.g. Fig 3).

Really nice with the PCA analysis and the figure. I suggest moving some of the labels (excellent with direct labeling) so that they are not overlapping.

I suspect there was more data available than 2 weeks per person. How were the 2 weeks chosen? Randomly?

Why didn’t the authors tune the XGBoost model?

---

## [Decision Letter · Decision Letter 1]

27 Jan 2025

PDIG-D-24-00260R1Chronobiologically-Informed Features from CGM Data Provide Unique Information for XGBoost Prediction of Longer-Term Glycemic Dysregulation in 8,000 Individuals with Type-2 DiabetesPLOS Digital Health Dear Dr. Smarr, Thank you for submitting your manuscript to PLOS Digital Health. After careful consideration, we feel that it has merit but does not fully meet PLOS Digital Health's publication criteria as it currently stands. Therefore, we invite you to submit a revised version of the manuscript that addresses the points raised during the review process. Please submit your revised manuscript within 30 days Feb 26 2025 11:59PM. If you will need more time than this to complete your revisions, please reply to this message or contact the journal office at digitalhealth@plos.org. Please include the following items when submitting your revised manuscript:* A rebuttal letter that responds to each point raised by the editor and reviewer(s). You should upload this letter as a separate file labeled 'Response to Reviewers '. This file does not need to include responses to any formatting updates and technical items listed in the 'Journal Requirements' section below.* A marked-up copy of your manuscript that highlights changes made to the original version. You should upload this as a separate file labeled 'Revised Manuscript with Track Changes '.* An unmarked version of your revised paper without tracked changes. You should upload this as a separate file labeled 'Manuscript '. If you would like to make changes to your financial disclosure, competing interests statement, or data availability statement, please make these updates within the submission form at the time of resubmission. Guidelines for resubmitting your figure files are available below the reviewer comments at the end of this letter. We look forward to receiving your revised manuscript. Kind regards, Hisham HasanGuest EditorPLOS Digital Health Leo Anthony CeliEditor-in-ChiefPLOS Digital Healthorcid.org/0000-0001-6712-6626 **Additional Editor Comments (if provided):** Thank you for the clarification. Upon reviewing the response again, I see that the issue regarding data availability has not been fully addressed. As per PLOS’s data availability policy, all underlying data must be made publicly available prior to acceptance, not just post-acceptance. This means that the data should either be included as part of the manuscript submission or deposited in a public repository with unrestricted access at the time of submission. Once this is resolved, I can proceed with the next steps in the review process.**Reviewers' Comments:**Review Comments to the Author

Reviewer #2: The authors have responded to my comments, carefully. No further suggestions. I think that the paper can be published as it is without any other request.

Reviewer #3: The authors have conducted a thorough revision, which improved the quality of the paper. I have two remaining comments.

(1) I find the new supplementary figures very informative, but they are not mentioned in the manuscript, so readers' attention is not called to them. I suggest to include them, especially the one on calibration, because the new features seem to improve this aspect.

(2) The authors state that both (some) de-identified data and code are available in the University of California, San Diego’s Open Data Repository. Please provide links (or doi) to help readers navigate to this useful resource.

---

## [Editor Report · Decision Letter 2]

5 Mar 2025

Chronobiologically-Informed Features from CGM Data Provide Unique Information for XGBoost Prediction of Longer-Term Glycemic Dysregulation in 8,000 Individuals with Type-2 Diabetes

PDIG-D-24-00260R2

Dear Asst. Prof. Smarr,

We are pleased to inform you that your manuscript 'Chronobiologically-Informed Features from CGM Data Provide Unique Information for XGBoost Prediction of Longer-Term Glycemic Dysregulation in 8,000 Individuals with Type-2 Diabetes' has been provisionally accepted for publication in PLOS Digital Health.

Best regards,

Hisham E. Hasan

Guest Editor

PLOS Digital Health